# RESIDUAL NETWORKS CLASSIFY INPUTS BASED ON THEIR NEURAL TRANSIENT DYNAMICS

## ABSTRACT

In this study, we analyze the input-output behavior of residual networks from a dynamical system point of view by disentangling the residual dynamics from the output activities before the classification stage. For a network with simple skip connections between every successive layer, and for logistic activation function, and shared weights between layers, we show analytically that there is a cooperation and competition dynamics between residuals corresponding to each input dimension. Interpreting these kind of networks as nonlinear filters, the steady state value of the residuals in the case of attractor networks are indicative of the common features between different input dimensions that the network has observed during training, and has encoded in those components. In cases where residuals do not converge to an attractor state, their internal dynamics are separable for each input class, and the network can reliably approximate the output. We bring analytical and empirical evidence that residual networks classify inputs based on the integration of the transient dynamics of the residuals, and will show how the network responds to input perturbations. We compare the network dynamics for a ResNet and a Multi-Layer Perceptron and show that the internal dynamics, and the noise evolution are fundamentally different in these networks, and ResNets are more robust to noisy inputs. Based on these findings, we also develop a new method to adjust the depth for residual networks during training. As it turns out, after pruning the depth of a ResNet using this algorithm,the network is still capable of classifying inputs with a high accuracy.

## 1 INTRODUCTION

Residual networks (ResNets), first introduced in (He et al., 2016), have been more successful in classification tasks in comparison with many other standard methods. This success is attributed to the skip connections between layers that facilitate the propagation of the gradient throughout the network, and in practice allow very deep networks to undergo a successful training. Apart from mitigating the gradient problem in deep networks, the skip connections introduce a dependency between variables in different layers that can be seen as a system state. This novelty provides an opportunity for interesting theoretical analysis of their functioning, and has been the underlying pillar for some interesting analysis of such networks from dynamical system point of view (Ciccone et al., 2018; Chang et al., 2018; Haber & Ruthotto, 2017; Lu et al., 2017; Liao & Poggio, 2016; Ruthotto & Haber, 2018; Chaudhari et al., 2017).

This study mainly emphasizes the importance of internal transient dynamics in ResNets on classification performance. Using a dynamical system approach, for a general ResNet, we derive dynamics of state evolutions of the residuals in different layers. It is well-known that very deep residual networks with weight sharing are equivalent to shallow recurrent neural networks, with similar performance to ResNets with variable weights between layers (Liao & Poggio, 2016). Inspired by this work, we study ResNets with shared weights and sigmoid activation functions, which provide a more tractable mathematical analysis. Then, we show empirically that those dynamics are observed in ResNets with variable weights as well. To study the classification performance of ResNets on noisy inputs, we compare the dynamics of noise evolution in these networks with that of Multi-Layer Perceptrons (MLP), and find that for small noise amplitudes, MLP has a higher value of signal to noise ratio. However, for increased noise amplitude, ResNets are more robust.

The main contributions of the paper are:

1. We show that network internal dynamics that are shaped by neural transient dynamics are well separated for each input class.

2. We compare the transient dynamics of MLP and ResNets, and show how a noise term in the input evolves in the network, and how it affects the classification robustness.

3. Based on residual dynamics, we develop a new method to obtain an adaptive depth for ResNets, during training, for input classification.

## 2  RELATED WORK

In (Haber & Ruthotto, 2017), the authors have studied residual networks using difference equations, and analyzed the stability of the forward propagation of input data, and have linked the inverse problem to the well-posedness of the learning problem. To circumvent the vanishing or exploding gradient problem, it is suggested in (Haber & Ruthotto, 2017) to design the eigenvalues of the feedforward propagation close to the edge of stability, so that the inverse problem is not ill-posed. It is however not clear whether having the eigenvalues set close to the edge of stability is beneficial for the network performance, because it depends on the dynamics required by the actual task. Employing this idea, the authors in (Chang et al., 2018) suggest a new reversible architecture for neural networks based on Hamiltonian systems. Also, in a recent study, ResNets have been employed as an unrolled non-autonomous time-invariant (with weight sharing) system of differential equations (Ciccone et al., 2018), wherein, each ResNet block receives an external input, which depends on the previous block. This successive process of feeding the following block by the output of the previous block continues until the latent space converges. Our approach in this paper is similar to the aforementioned studies, however, to understand the classification mechanism in ResNets, we focus on the role of the intrinsic transient dynamics of the residuals over different layers of the network.

Some studies on ResNets have focused on tracking the features layer by layer (Greff et al., 2017; Chu et al., 2017), and have challenged the idea that deeper layers in neural networks build up abstract features that are different than those formed in lower layers. One supporting evidence for this challenge comes from lesion studies on ResNets (Veit et al., 2016) and Highway networks (Srivastava et al., 2015) which show that after the network is trained, perturbing the weights in the deep layers does not have a fundamental effect on the network performance, and therefore, does not bring the performance to chance level. However, changing the weights which are closer to initial layers, have more damaging effect. Empirical studies in (Greff et al., 2017; Chu et al., 2017) suggest an alternative explanation for feature formation in deep layers; that is, successive layers estimate the same features which, along the depth of the network, are more refined, and yield an estimate with smaller standard deviation than earlier layers. Our study supports this idea by showing that features in different layers of a ResNet with shared weights are formed by the transient dynamics of residuals that may converge towards their steady state values if they are stable. In attractor networks, perturbing the initial layers changes those dynamics more drastically compared to perturbation of deeper layers, because the residuals in the deep layers are either very close to their stable fixed point, or have already converged. If there are no attractors for the residuals, sensitivity to initial conditions and the internal dynamics of the residuals play an important role in classification. In this case, perturbations of the network at initial layers can potentially change the dynamic evolution of the residuals completely, and this will have a more sever impact on the output. Classification based on unstable internal states is similar to Reservoir networks (Maass et al., 2002), where it has been shown that the high dimensionality of the neurons at the readout layer can compensate for the lack of stability of the neural activities.

Moreover, one important topic in this domain is the depth of ResNets. On the one hand, the success of ResNets in classification has been attributed to their deep architecture (He et al., 2016), on the other hand, there are studies that claim most of the training is accomplished in the initial layers, and having a very deep architecture is not necessary (Zagoruyko & Komodakis, 2016). Another challenge is to understand the generalization property of ResNets (which may be related to its depth), because their power is correlated with their ability in recognizing unseen data that also belong to the classes that these networks have been trained on. In our analysis of the transient and steady state dynamics of the residuals, we discuss these issues. In fact, an important finding of this paper is that residual networks classify inputs based on summing over all the residual's outputs throughout the network, meaning different transitions of residuals (convergence to their steady state, or their long wandering

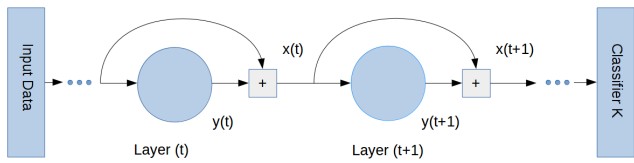

Figure 1: A simple schematic of skip connections between two successive layers. The dimensionality of the network does not change with depth. The variable $\mathbf{y}(t)$ represents the values of the residuals at layer $t$, and $\mathbf{x}(t)$ is the activation of neurons at layer $t$.

trajectories without convergence) can potentially change the classification result. Interestingly, in biological neuronal networks, it has been suggested that optimal stimulus separation in neurons that encode sensory information occurs during the transients rather than the fixed points of the neuronal activity trajectories (Mazor & Laurent, 2005; Rabinovich et al., 2008). Also, it has been discussed that spatio-temporal processing in cortical circuits are state dependent, and the role of transients are crucial (Buonomano & Maass, 2009). In our study we show that also in ResNets, these transients are the decisive factors for classification. Based on this finding, we show how a noise term in the input is mapped to the output, and we develop a new method to control the depth of ResNets.

## 3 DYNAMICS OF INTERACTIONS BETWEEN RESIDUALS

We consider a dense ResNet with $N$ input dimensions, and arbitrary $T$ layers with exactly $N$ neurons at each layer. A unique property of a ResNet that distinguishes it from conventional feedforward networks (such as MLP) is the skip connection between layers. In the ResNet we consider here, the activity of neuron $i$ at layer $t$, before the output of the previous layer $(t-1)$ is added to it, is represented as $y_i(t)$, and the activity of all neurons in the same layer is represented by the vector $\mathbf{y}(t)$. After the integration of the output from layer $t-1$, the output of layer $t$ is represented by $\mathbf{x}(t)$. The components of these residuals $\mathbf{y}(t)$ are calculated based on a linear function of $\mathbf{x}(t)$, i.e. $z_i(t) = \sum_{i=1}^{N} w_{ij}(t)x_j(t)$ followed by a nonlinear function $f(z_i)$. Figure 1 illustrates the relation between $\mathbf{x}$ and $\mathbf{y}$. Any hidden layer $t$ represents a sample of the dynamical states $\mathbf{x}$ after $t$ steps. Input data is considered as the initial condition of the system, and is depicted by $\mathbf{x}(0)$. Interpreting the network as a dynamical system which evolves throughout the layers, the dynamics of neural activations are $\mathbf{x}(t+1) = \mathbf{x}(t) + \mathbf{y}(t+1)$, where $\mathbf{y}(t)$ is the output of the neurons, and in the rest of the paper, they are called "residuals". This equation implies a difference equation for the variable $\mathbf{x}(t)$, that is $\mathbf{x}(t+1) - \mathbf{x}(t) = \mathbf{y}(t+1)$. The left side of this equation resembles the forward Euler method of derivative of a continuous system, when the discretization step is equal to 1. This approximates a continuous system with dynamics that follow $\dot{x}_i(t) = y_i(t)$.

We are interested in understanding how the neural activities evolve over layers in a feedforward fully connected ResNet. The dynamics of $\mathbf{x}(t)$ and inputs to the residuals, $\mathbf{z}(t)$, follow

$$\dot{x}_i(t) = y_i(t) = f(z_i(t)) \Longrightarrow x_i(t) = \int_0^t y_i(\tau)d\tau + x_i(0)$$

$$z_i(t) = \sum_{j=1}^{N} w_{ij}(t)x_j(t) + b_i \Longrightarrow \dot{z}_i(t) = \sum_{j=1}^{N} w_{ij}(t)\dot{x}_j(t) + \dot{w}_{ij}(t)x_j(t) \tag{1}$$

The first line of equation 1 indicates that $\mathbf{x}(t)$ stores the sum of the residuals and the input ($\mathbf{x}(0)$) from the input layer up to layer $t$, meaning that $\mathbf{x}(t)$ is a cumulative signal for $\mathbf{y}(t)$, as well as the input data. For a ResNet with shared weights between layers, $\dot{w}_{ij}(t) = 0$; because the weights do not change between layers. This constraint makes the analysis simpler, and results in $\dot{z}_i(t) = \sum_{j=1}^{N} w_{ij}\dot{x}_j(t)$. In this case, after replacing $\dot{x}_i(t)$ by $y_i(t)$ in equation 1, the dynamics of $\mathbf{z}(t)$ and the residuals $\mathbf{y}(t)$

will be

$$\dot{z}_i(t) = \sum_{j=1}^{N} w_{ij}(t)y_j(t)$$

$$\dot{y}_i(t) = \frac{\partial y_i(t)}{\partial z_i(t)} \frac{\partial z_i(t)}{\partial t} = \frac{\partial f(z)}{\partial z}(\sum_{j=1}^{N} w_{ij}(t)y_j(t)). \tag{2}$$

A steady state solution for $y_i$, in a network with a time invariant weight $w_{ij}$ (shared weights across layers), is obtained from setting either of the two terms on the right hand side of equation 2 to zero. For a general nonlinear $f(.)$, according to equation 2, the interactions between the residuals are either competitive or cooperative, depending on the positive or negative influence that they have on each other. For a *logistic* function $f(.)$, the derivative is $y_i(t)(1 - y_i(t))$, which results in a particular form of predator-prey equation, well-known in studying ecosystems. In this case, equation 2 yields

$$\dot{y}_i(t) = y_i(t)(1 - y_i(t))(\sum_{j=1}^{N} w_{ij}(t)y_j(t)) \tag{3}$$

For $N$ residuals, the number of possible fixed points (solutions of equation 3) is $3^N$. However, for a given weight $W$, only some of the fixed points are stable. The initial condition for the residuals is determined by the output of the first layer. After some transients over the next layers, each residual converges to its stable solution, if there is one. Note that the derivatives at $y_i = 0$ or 1 are equal to zero, and the system's trajectories are confined to this space. According to equation 1 and figure 1, the cumulative of the residuals over the entire network feeds the classifier. Therefore, perturbations at initial layers of the network disrupt the output more severely than perturbations of the deeper layers (lesion studies have considered weight perturbations, which reflects on neural activity perturbations). A discrete-time analysis of the effect of perturbations at different layers on the output value is given in the Appendix. To study the contribution of input noise ($\xi$) in the final hidden layer, we derive the noise evolution for the ResNet considered above (a similar analysis applies to ResNets with continuous activation functions). Represented in vector notations, the signal plus noise evolution is obtained by replacing $y_i$ in equation equation 3 by $\mathbf{y} + \xi$.

$$\begin{aligned} \dot{\mathbf{y}} + \dot{\xi} &= (\mathbf{y} + \xi)(1 - \mathbf{y} - \xi)(W(\mathbf{y} + \xi)) \\ &= \mathbf{y}(1 - \mathbf{y})(W\mathbf{y}) + \mathbf{y}(1 - \mathbf{y})(W\xi) + \xi(1 - 2\mathbf{y})(W(\mathbf{y} + \xi)) - \xi^2(W(\mathbf{y} + \xi)) \end{aligned} \tag{4}$$

Assuming that $\mathbf{y(t)}$ is the solution of the neural activity at layer $t$ for the input signal without noise (that is reflected in initial condition $\mathbf{y}(1)$), subtracting equation 3 from equation 4 yields

$$\begin{aligned} \dot{\xi} &= \mathbf{y}(1 - \mathbf{y})(W\xi) + \xi(1 - 2\mathbf{y})(W(\mathbf{y} + \xi)) - \xi^2(W(\mathbf{y} + \xi)) \\ &= \mathbf{y}(1 - \mathbf{y})(W(\xi + \mathbf{y} - \mathbf{y})) + \xi(1 - 2\mathbf{y} - \xi)(W(\mathbf{y} + \xi)) \\ &= \mathbf{y}(1 - \mathbf{y})(W\mathbf{y}) + \mathbf{y}(1 - \mathbf{y})(W(\xi - \mathbf{y})) + \xi(1 - 2\mathbf{y} - \xi)(W(\mathbf{y} + \xi)) \\ &= \dot{\mathbf{y}} + \mathbf{y}(1 - \mathbf{y})(W(\xi - \mathbf{y})) + \xi(1 - \mathbf{y} - (\mathbf{y} + \xi))(W(\mathbf{y} + \xi)) \end{aligned} \tag{5}$$

In equation 5 and equation 4, the vectors $\mathbf{y}$ and $\xi$ are functions of $t$. This equation indicates how the growth rate of the noise ($\dot{\xi}$) differs from the growth rate of the residuals ($\dot{\mathbf{y}}$).

## 4 NEURAL DYNAMICS IN MLP

An MLP can also be interpreted as a dynamical system. Given that in a feedforward MLP, $y(t + 1) = f(z)$, to derive the neural state dynamics, a $-y(t)$ term can be added to both sides of this equation. Using the same argument as in ResNets, the internal dynamics of MLP can be written as

$$\dot{y}_i(t) = -y_i(t) + f(z(t)) = -y_i(t) + \frac{1}{1 + \exp(-W\mathbf{y})}$$

$$\dot{\mathbf{y}} + \dot{\xi} = -\mathbf{y} - \xi + \frac{1}{1 + \exp(-W(\mathbf{y} + \xi))} \tag{6}$$

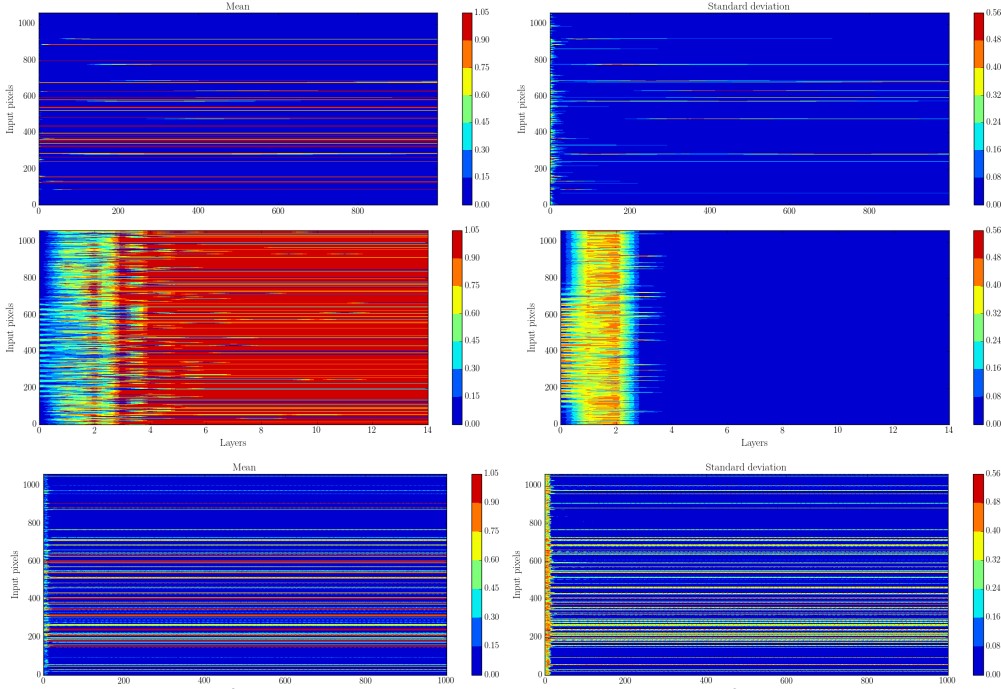

Figure 2: Mean and standard deviation of the residuals for all 10 classes for a 1000 layer deep ResNet with shared weights (top), a 15 layer ResNet with variable weights (middle), and mean and standard deviation of the residuals of the MLP network for inputs corresponding to 2000 samples from all classes. Top: The weights are from a 15-layer deep network. The stable residuals are sparse in activities. Middle: The network with 15 different weight matrices has more residuals at their maximum values which correspond to many saturated neurons in the final layer. The standard deviation of the residuals are zero after the 4th layer, showing that the residuals have no transient dynamics in the following layers. Bottom: Residual dynamics in MLP are oscillatory and the standard deviation between residuals corresponding to a single input class is high.

For a small noise, the last term in the equation above can be approximated by a Taylor expansion. Keeping only up to first order terms of this expression results in

$$
\begin{aligned}
\dot{\mathbf{y}} + \dot{\xi} &= -\mathbf{y} - \xi + \frac{1}{1 + \exp(-W(\mathbf{y} + \xi))} + \xi(\frac{1}{1 + \exp(-W\mathbf{y})})(1 - \frac{1}{1 + \exp(-W\mathbf{y})}) \\
\dot{\xi} &\simeq -\xi + \xi(\dot{\mathbf{y}} + \mathbf{y}) - \xi(\dot{\mathbf{y}} + \mathbf{y})^2 \\
&\simeq \xi(\dot{\mathbf{y}} + \mathbf{y} - (\dot{\mathbf{y}} + \mathbf{y})^2 - 1)
\end{aligned}
\tag{7}
$$

The last equation indicates that for small noise amplitudes, the noise is always suppressed as it propagates in the network. The reason is that the elements of $\dot{\mathbf{y}} + \mathbf{y}$ are always negative (property of the sigmoid function). However, for larger noise amplitudes, higher order terms in the Taylor approximation will be needed. This fact is nicely reflected in the noise to signal ratio of the MLP network for small noise perturbations in figure 4. Including the second order perturbation terms to the Taylor expansion, results in

$$
\dot{\xi} \simeq -\xi + \xi(\dot{\mathbf{y}} + \mathbf{y})(1 - \dot{\mathbf{y}} - \mathbf{y})(1 + \frac{\xi}{2}(1 - 2\mathbf{y} - 2\dot{\mathbf{y}}))
\tag{8}
$$

## 5 EXPERIMENTS ON MNIST

To study the behavior of the network on large datasets, we considered a network with 1064 sigmoid neurons in each layer, and 15 layer deep. First, we analyzed a case where the weight matrix was

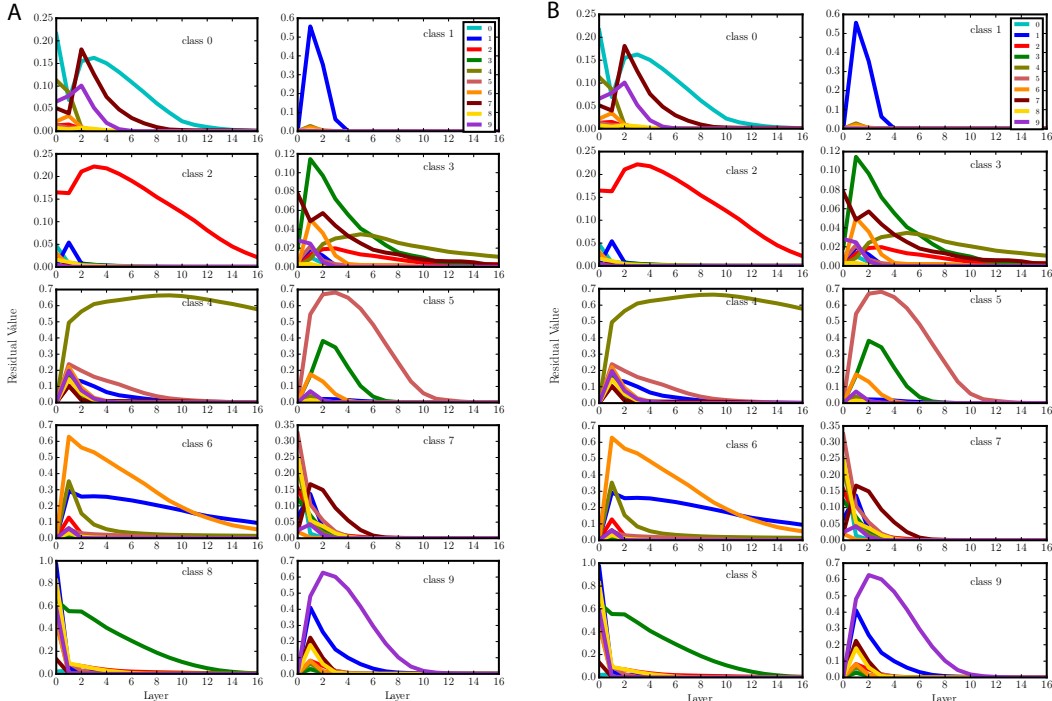

Figure 3: Residuals (A) and their cumulative (B) for neurons that have the largest contribution in the classifier's output for the softmax layer. In all cases, but class 8, the cumulative with the highest final value at layer 15 corresponds to the class that has the highest sensitivity to that neuron.

shared between all the layers. The input data was chosen from MNIST, and the classification was performed by using a softmax function. In this case, the classification error on the test set was $1.4\%$. Note that the network considered here is the simplest possible network architecture (so as to allow us to understand the classification mechanism in ResNets) with shared weights; therefore, the results are not comparable to the state of the art performance on MNIST. As illustrated in figure 2 top panel, the residuals in the first few layers are still in the transition period (non-zero standard deviation for 200 random samples from 10 classes). We used the same weights in a network with 1000 hidden layers (without retraining) to study the behavior of the residuals in a deep realization of the same ResNet. There were a few non-zero fixed points with some negligible standard deviation among 200 samples. To check if the fixed points were different and distinguished for each class, we plotted the average and the standard deviation of the residuals for the test set, separately for each class, on the final hidden layer in figure 5 in the appendix. The average for each class is different from any other class, however, a large number of dimensions are identical. The standard deviation between the residuals in a single class are small, and negligible, and indicate that the classification accuracy is not $100\%$. For this prolonged simulation, we obtained the eigenvalue distribution for the average residuals for each class at layer 1000. Residuals corresponding to classes $0, 2, 3$ had a single small positive eigenvalue (around 0.02) among all other negative eigenvalues (indication of the existence of saddle point). This means that those classes are still in their transition period at layer 1000, and due to the small value of the positive eigenvalue, the transition is slow.

Due to the high dimensionality of the network, it is not viable to illustrate the transition dynamics of individual neurons separately for each class. However, to show different transition patterns of the residuals in each class, we chose one neuron for each class such that the classifier had the highest sensitivity to the value of the cumulative transitions of that neuron. The index of this neuron was derived from the sensitivity of each class $C$ with respect to $\mathbf{x}(T)$, which is the classifier $K$. For each output in the softmax layer, there exists a maximally sensitive weight for its corresponding classifier $K$. This method renders 10 different indices. In figure 3A, we plotted the average (over 1000 samples for each class) of the residuals for neurons that corresponded to those indices. In almost all cases (apart from class 8), the maximum value of $x(15)$ belonged to the neuron that had

the largest coefficient in the classifier vector for that particular class. This implies that separation between classes are encoded in the transient dynamics of those neurons (and other neurons that their cumulative trajectories are multiplied by big coefficients in the classifier). The transient dynamics of these neurons play an important role in the classification result. To show that these transient dynamics are different and separable for each input class, we projected the trajectories of the residuals to a 2D plane, and observed a clear separation between the internal dynamics for each class (Appendix C).

In a different experiment, we investigated the behavior of a similar ResNet, with 15 layers, but with variable weight matrix for each layer. The mean and standard deviations of the residual for 200 samples are illustrated in figure 2, bottom panel. In this case, the residuals converge to their steady state solutions already on the fourth layer, as their standard deviations across samples converge to zero after the fourth layer. A striking finding in this case is that the standard deviation of the residuals for samples from different classes are zero, meaning that only one stable fixed point encodes all the similarities between different input classes. Considering this fact, we conclude that the sum of transient dynamics across layers for different input classes converges to different outputs that discriminate the inputs. Another interesting observation is that at the few layers close to the output layer, the weight matrix between layers converged to a fixed matrix. Also, compared to the previous example of a network with weight sharing, there are more residuals that converge to nonzero values. This gives the network enough capacity to give divergent outputs for different classes, based on their initial conditions. In this example, the classification error on the test set was about $1.8\%$. This higher value of the error rate might be due the paucity of separate fixed points to represent each class. Considering the observation that a ResNet with multiple fixed points for the residuals, corresponding to different classes, renders a smaller classification error, hints to the point that having different similarity representations of the input encoded in the residuals results in a better generalization, compared to cases where only one single fixed point for residuals stands for the entire input classes.

For the MLP network applied on MNIST dataset, the residuals show an oscillatory dynamics, and no convergence at layer $1000$ was observed (figure 2 bottom panel).

To compare the robustness of the ResNet and MLP network to noise in the input data, we injected a uniformly distributed noise $\xi$ of different amplitude to the initial conditions. We calculated the noise to signal ratio ($\sum_{\mathbf{t=0}}^{\mathbf{T}} \xi(\mathbf{t})/\mathbf{x}(T)$) for the ResNet, and ($\xi(T)/\mathbf{y}(T)$) for the MLP network, for different classes with $10$ independent realizations of the input noise. The average noise to signal ratio is depicted in figure 4A. For a maximum noise amplitude less than $0.1$, the noise to signal ratio in the MLP network is smaller than that of the ResNet, which means that MLP is able to suppress the noise term better, and will render a bigger cosine similarity in the last hidden layer. However, for a maximum noise amplitude in the range of $0.1$ and $1.$, the MLP fails to cancel out the noise term, and the noise to signal ratio grows quickly as a function of noise amplitude. Already at $0.1$ maximum noise amplitude, the network misclassifies the noisy input signals. The ResNet, however, shows a higher noise to signal ratio for small amplitudes of noise, and the misclassification occurs at maximum amplitudes that are larger than $0.5$. This implies more robustness of ResNets to input perturbations. As a suggestion to increase the robustness of ResNets even further, it is possible to add a regularization term to the cost function of the network which includes the integration of the right hand side of equation 5 for the last two hidden layers (this could be applied on the weights of the last hidden layer in a general network with variable weights, and with a continuous activation function.)

## 6    ADAPTIVE DEPTH FOR RESNETS DURING TRAINING

Results of the previous sections shed some light on the mechanism of classification in ResNets. After understanding the role of transient dynamics in input classification, we envisage a new method to design the depth of ResNets based on the layer-dependent behavior of the residuals. In this method, during training, initially an arbitrary number of layers is chosen. After training each epoch using the back-propagation algorithm, the difference between the residuals for the last successive layers of the ResNet block are calculated. If this difference is less than a minimum threshold (we chose $0.01$ for each neuron on average), the last hidden layer in the block is to be removed, because the value of the residuals will not contribute much to the cumulative function. This process continues until the network is trained (minimum loss on the training set). Note that convergence of the residuals is not a necessary requirement for classification, but a sufficient condition; i.e. when the residuals converge, and when the training loss is minimum, there is no need for extra layers in the blocks (and also before

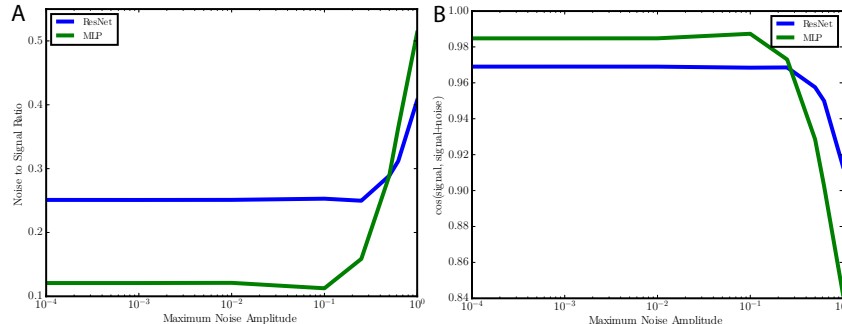

Figure 4: Average noise to signal ratio (A) and cosine similarity (B) for final hidden layers of ResNet, and MLP network. The horizontal axis is in logarithmic scale, and shows the maximum amplitude of the noise signal.

the classification layer). This algorithm can be implemented as a piece of code in parallel with other training algorithms for ResNets:

**while** *loss function is not minimum* **do**
    **for** *each epoch of the training data* **do**
        residuals of the last hidden layer in the block $\rightarrow r_1$
        residuals of the second last hidden layer in the block $\rightarrow r_2$
        **if** $l_1$ *norm for* $r_1 - r_2 < threshold$ **then**
            remove the layer corresponding to $r_1$
        **end**
    **end**
**end**

In the examples shown in the previous sections, we demonstrated a converging behavior of the residuals to stable or metastable (saddle fixed points) states. For a fully connected network with variable weights between layers, we applied our algorithm during training on MNIST dataset. The algorithm was able to shrink the network to 3 layers, and the error on the test set was $1.7\%$ (similar to the network performance with $15$ layers). Also, for a network with shared weights between layers, the network depth was reduced to $5$ layers without any changes in classification accuracy.

## 7 CONCLUSION

In this study, we showed that given an input, ResNets integrate samples of the residuals from each layer, and build an output representation for the input data in the final hidden layer. This sum depends on the initial condition (input data) and its transition towards the steady state of the corresponding residual. In some networks which show attracting and converging behavior, one or more stable fixed point for the residuals exists. In other cases, among many other possible dynamics, multiple fixed points for different input classes might exist, some of which could be stable or metastable. In both cases, different neural transient dynamics (with inputs of different classes as initial conditions) can result in different cumulative values of the residuals, and therefore, different classification outcomes. Using a dynamical system interpretation of the networks, we compared internal dynamics of an MLP network with that of a ResNet on MNIST dataset, and we showed ResNets are more robust to signal perturbations. We also developed a new method for designing an adaptive depth for ResNets during training. The main idea is that after all the residuals have settled into their steady state value, or if there are negligible changes of the values of the residuals between successive layers, there is no need for any extra deeper layers. This is because any additional layer of the residual neurons will add almost the same values as the previous layer, without any extra information about the neural transitions.

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

## 8 APPENDIX

### 8.1 A: SENSITIVITY ANALYSIS OF THE OUTPUT WITH RESPECT TO LAYER PERTURBATIONS

As mentioned in the Introduction, some lesion studies have shown that weight perturbations at the initial layers of the network can have more sever consequences on output classification results than perturbations of the weights at deeper layers. Since we have considered a network with shared weights,we study the effect of perturbations of the residuals (which could be considered as the result of weight perturbations in previous studies). To understand how slight perturbations of the residuals affects the values of the output, we analyzed the sensitivity of the output $C$ with respect to slight perturbations of the residuals $\mathbf{y}(t)$. Assuming slight perturbations on $\mathbf{y}(s)$, we are interested in the evolution of this perturbation throughout the network and its effect on the output unit. We used the chain rule of differentiation on the discrete time dynamics of the network to obtain a sensitivity matrix that propagates the perturbations from $\mathbf{y}(t)$, layer to layer, until it reaches the output $\mathbf{C}$ (the analysis is in discrete case). The sensitivity equation reads

$$S(t) = \frac{\partial \mathbf{C}}{\partial \mathbf{y}(s)} = \frac{\partial \mathbf{C}}{\partial \mathbf{x}(t)} \frac{\partial \mathbf{x}(t)}{\partial \mathbf{x}(t-1)} \frac{\partial \mathbf{x}(t-1)}{\partial \mathbf{x}(t-2)} \cdots \frac{\partial \mathbf{x}(s)}{\partial \mathbf{y}(s)} \qquad (9)$$

where $t = T$ is the last hidden layer. Based on the definition of $\mathbf{x}(t)$, it is easy to verify that $\frac{\partial \mathbf{x}(t)}{\partial \mathbf{x}(t-1)} = \mathbf{I} + \frac{\partial \mathbf{y}(t)}{\partial \mathbf{x}(t-1)}$. We represent $\frac{\partial \mathbf{y}(t)}{\partial \mathbf{x}(t-1)}$ by $M(t)$, which is $\frac{\partial \mathbf{f}(W\mathbf{x}(t-1))}{\partial \mathbf{x}(t-1)}$. Using vector representations, $M = \mathbf{y}(t) \odot (1 - \mathbf{y}(t)) \odot W$, where $\odot$ represents element-wise multiplications. Since the output class $\mathbf{C}$ is the result of the inner product between the classifier vector $K$ and the cumulative signal $\mathbf{x}(T)$, $\frac{\partial \mathbf{C}}{\partial \mathbf{x}(T)} = K$. It is also clear that the last term on the right hand side of equation 9 is equal to the identity matrix. Therefore, in equation 9, the sensitivity matrix can be represented as

$$S(t) = K[I + M(T)][I + M(T-1)][I + M(T-2)] \cdots [I + M(s)] = KM^{'}(s) \qquad (10)$$

For different layers $s_1$ and $s_2$, where $s_2 > s_1$, we calculated $M^{'}(s_1)$ and $M^{'}(s_2)$. It turns out that for network simulations that we performed (see Experiment section), for $s_2 = 10$ and $s_1 = 4$, the determinant of $M^{'}(s_1)$ is orders of magnitude larger than the determinant of $M^{'}(s_2)$. This implies perturbations at initial layers are greatly amplified at the final hidden layer compared to perturbations at close to final layers. Since the final hidden layer is multiplied by the classifier vector $K$, those amplified perturbations will have a more disruptive consequence on the output of the network.

### 8.2 B: LONG-TIME BEHAVIOR OF THE RESIDUALS ON MNIST DATASET

Initially, we trained a ResNet with 15 layers with shared weights between layers. To observe the long-time behavior of the residuals in this network, we used the same weights in a network with 1000 hidden layers (without retraining). There were a few non-zero fixed points with some negligible standard deviation among 200 samples. To check if the fixed points were different and distinguished for each class, we plotted the average and the standard deviation of the residuals for the test set, separately for each class, on the final hidden layer in figure 5. The average for each class is different from any other class, however, a large number of dimensions are identical. The fact that each class has a distinguished fixed point indicates that the trajectories of the residuals for each class are separated. According to figure 5, for the ResNet, the standard deviation between the residuals in a single class are small, and negligible, and indicate that the classification accuracy is not $100\%$. For the MLP network, the representations at layer 1000 are more diverse, and the standard deviations of this state for different input classes is high. This reflects the fact that the activities of the neurons in this network did not converge to a fixed value (in fact, the neural dynamics are oscillatory).

### 8.3 C: INTERNAL REPRESENTATIONS FOR RESNET AND MLP

To illustrate the role of internal transient dynamics in input classification, we mapped the residual signals from a 1064 dimensional state space to a 2 dimensional distance space, using the Umap algorithm Mcinnes & Healy (2018). This algorithm provides a dimensionality reduction technique based on Riemannian geometry which preserves more of the global structure, and has a superior run time performance compared to t−SNE Maaten & Hinton (2008). As shown in figure 6A, the

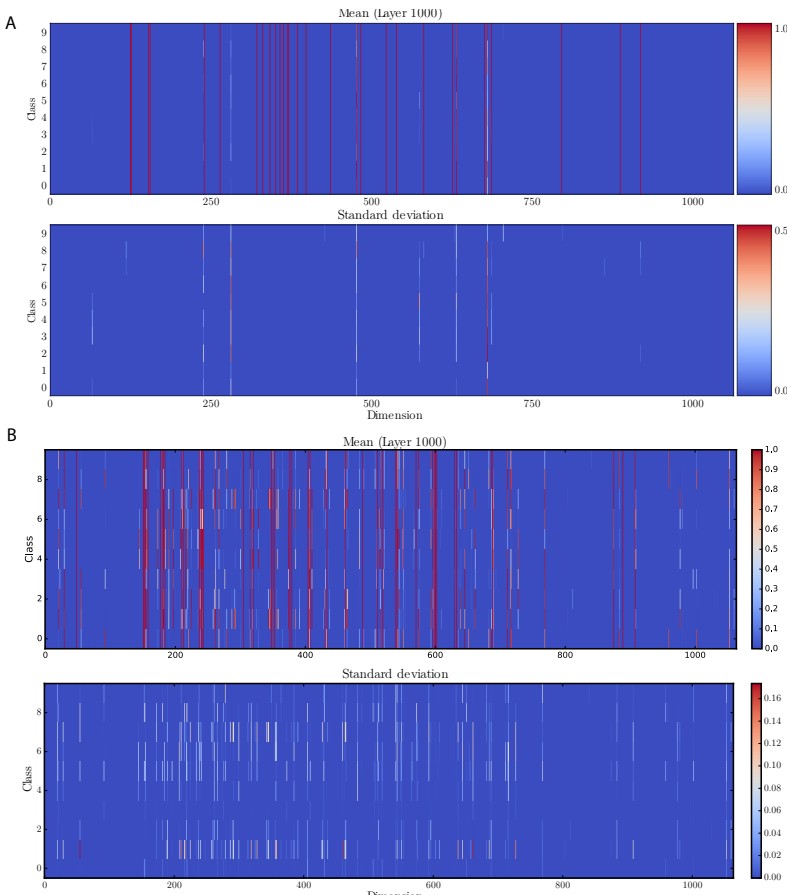

Figure 5: Mean and standard deviation of the residuals, for the ResNet with shared weights (A), and the MLP network (B), corresponding to the 10 different classes of the MNIST test data. The mean of the residuals are different in few dimensions for each class in the ResNet, and there are more variabilities in the MLP.

transient dynamics that correspond to different input classes build up clusters in a 2D space that are well separated from each other. A similar representation holds for reservoir networks that are used for analyzing time-dependent inputs Jaeger & Haas (2004); Buonomano & Maass (2009).

For the ResNet with shared weight between layers, we calculated the average cosine similarity between the final hidden layers of 200 samples per each class. The cosine similarity is shown in a symmetric matrix in figure 6B, where in each row $i$ the element in column $i$ has a higher values. This means more similarity between outputs of the last hidden layer for inputs of the same class. The same calculation was performed for the MLP network (figure 6C), and in this case, the cosine similarity for the last hidden representation of inputs from the same class is bigger, and the difference between classes is more pronounced than in the ResNet scenario.

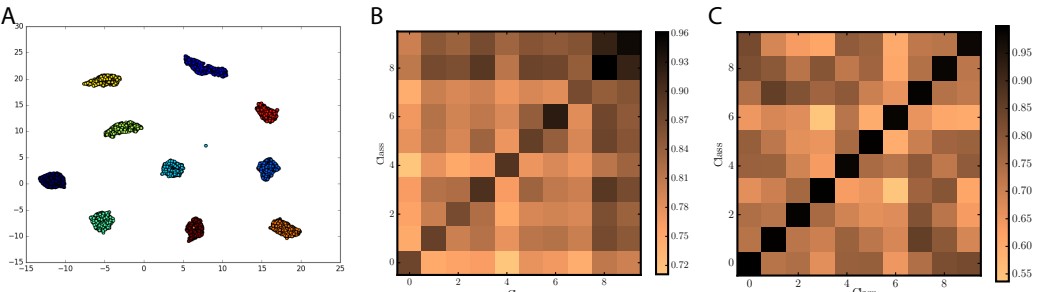

Figure 6: Clustering properties of internal dynamics for the ResNet (with shared weights) and the MLP network considered in the paper. A: distance between residuals corresponding to different classes. Residuals that belong to the same input class are closer to each other than those belonging to different classes. B: Average cosine similarity between the final hidden layers of the ResNet for 200 samples per class. C: Average cosine similarity between the final hidden layers of the MLP network for 200 samples per class.

