# OpenReview forum: "RESIDUAL NETWORKS CLASSIFY INPUTS BASED ON THEIR NEURAL TRANSIENT DYNAMICS"
_ICLR.cc/2019/Conference_

### Official Review · AnonReviewer2 · 2018-11-02
**Interesting direction, but questionable significance of contributions**

**Rating:** 5
**Confidence:** 5

**Review:**

This paper presents the following main insight (quoting the authors): Resnets classify input patterns based on the sum of the transient dynamics of the residuals in different layers.
The authors formulate this insight mathematically, and experimentally demonstrate its validity on a toy binary classification problem. They also show that the behaviour of a Residual network trained on MNIST is in line with the claims. Finally, a method to adapt the depth of a Residual network during training is proposed and applied to the toy classification problem.

The paper is generally of good quality and easy to understand. My only complaint: the introduction (including related work) is too long and I think it will be unclear to a general reader before reading section 2, where the terms used in the paper are explained and clarified. I think it will be an improvement to leave out detailed discussion of related work for a separate section, and focus on clarifying what the paper is about.

Overall, while the paper is related to an interesting and potentially fruitful perspective of neural networks (as dynamical systems), in my view the contributions are not significant at this stage. That the sum of transients determines network outputs is almost by design, and can be shown without a dynamical systems perspective. Using the paper’s notation, one can sum over the equations for all the layers to obtain this.

x(1) = x(0) + y(1)
…
x(T) = x(T-1) + y(T)
————————————————————————
x(T) = x(0) + sum(y(t))

Since the classification is performed using x(T), it is clear that the sum of transients is what causes the change in representation and that y(T) can be the same or not for different class inputs.

Based on my understanding, I don’t find the findings to be significantly novel or surprising. In particular, I don’t see any concrete benefits of employing a dynamical systems perspective here. Nevertheless, the experimental analysis is interesting, and I’m hopeful that this direction will lead to more insights in the future.

The final contribution is a new method to learn the depth of Residual networks during training, but this is insufficiently explored (only tested on the toy dataset) so its practical significance can not be evaluated at this stage.


Minor notes:
- Note that many citations are textual when they should be parenthetical.
- The reference “No & Liao (2016)” has incorrect author name.

---

> ### Author Response · Authors · 2018-11-27
> **Author response**
>
> We thank the reviewer for his/ her helpful comments.
>
> We changed the introduction, and we moved the majority of the text to the following subsection according to the reviewer’s suggestion.
>
> We agree that it is simple to realize that ResNets sum over the transient dynamics due to the skip connections, and that this does not require a dynamical system analysis. However, the point is that the internal dynamics of the neurons that contribute in this sum are fundamentally different compared with a network that does not have the skip connections. We derived those dynamics, and we aimed at showing separable internal states for each class. Also, using concepts of stability analysis, we came up with a pruning algorithm at the end of the paper. In the recent version of our manuscript, we have compared ResNets and MLPs and have added some analysis on the response of the network to noisy inputs, using the dynamical system framework.
>
> For the section on adaptive depth, we have added more results on MNIST dataset (for a ResNet with variable weights, as well as shared weights), and have compared the result of the algorithm with the results of the previous section (basically the accuracy does not change much).

---

### Official Review · AnonReviewer1 · 2018-11-02
**Review of "Residual Networks Classify Inputs Based On Their Neural Transient Dynamics"**

**Rating:** 2
**Confidence:** 4

**Review:**

This paper aims to view the computations performed by residual networks through the lens of transient dynamical systems. A rough intuition is provided, followed by experiments in a toy concentric circle example and on MNIST. Finally, a method to determine the appropriate depth of ResNets is proposed, though experiments are only performed in the toy concentric circle experiment.

The approach of attempting to interpret feedforward networks through a dynamical systems perspective is an interesting and worthwhile one. However, this paper suffers from a number of flaws, and is clearly not ready for publication.

The clarity of this paper can be significantly improved. In general, the text is confusing, and as currently written, it is difficult to understand the central narrative of the manuscript. The review of literature in the introduction is relatively complete, though again, the presentation makes this section difficult to understand.

Scientifically, it is clear that further experiments on less toy datasets and settings will be required. While MNIST is useful for prototyping, experiments on datasets such as CIFAR (or ideally ImageNet) will be necessary to evaluate whether the observations made hold in more realistic settings. Moreover, the primary claim: that ResNets sum the residuals across layers is by definition true and by design. The scientific contribution of this statement is therefore questionable.

In addition, the case analyzed in the majority of the paper -- weight sharing across all layers -- is an unusual one, and as Figure 3 shows, clearly changes the network dynamics. The use of sigmoid activation functions is also an unusual one given that ResNets are generally trained with ReLU activations.

Finally, the proposed method for determining the optimal depth for ResNets is an interesting idea, and worth further examination. However, the paper currently evaluates this only on an an extremely toy dataset of concentric circles. Evaluation on more realistic datasets (comparatively) with appropriate baselines will be required to determine whether this method is in fact helpful.

---

> ### Author Response · Authors · 2018-11-27
> **Author response**
>
> Thank you very much for your comments.
>
> We chose MNIST to be able to systematically study the transient dynamics in a network that does not change the dimensionality as a function of time. We have added more analysis in the new version of the paper. Using more realistic datasets is a subject of our future work.
>
> In fact, ResNets were not originally designed to sum up the transient dynamics, but to facilitate back-propagation in a very deep network. Introducing the skip connections, however, as a byproduct, changes the neural dynamics completely. Neurons at each layer receive an input which carries the history of the neural responses, and this fundamentally changes the network dynamics. We have compared ResNets with multi layer perceptrons to make this point more clear.
>
> We chose sigmoid functions due to their continuous function which gives a continuous derivative as well. Also they have a bounded output property which helps with the analysis. Shared weights were chosen so that we have a time-invariant system, and could study the stability of the network (such as the number of stable fixed points). However, we have a figure for a more general case with variable weights as well, and we compared this case with the shared weight network.
>
> In the new version of the paper, the experiments for the final section are done for MNIST dataset, on the same networks that we studied in the previous sections, and the results are compared with the performance of the original deep networks. There was no significant changes in the classification accuracy after applying our algorithm, however, we observed a huge reduction of the number of layers. We will apply this algorithm on more realistic datasets.

---

> > ### Comment · AnonReviewer1 · 2018-11-27
> > **Response to rebuttal**
> >
> > Thank you for your response. While I appreciate that the various simplifications used in this paper (MNIST only, sigmoids, shared weights, etc.) make the analysis easier, they also reduce the likelihood that results found in these regimes will generalize to more realistic models and domains.
> >
> > Additionally, while the motivation for ResNets was indeed to facilitate back-propagation, the mathematical byproduct of this is indeed that ResNets sum residuals. This is explicitly stated in some of the initial ResNet papers. e.g.:
> >
> > "The feature x_L...of any deep unit L, is the summation of the outputs of all preceding residual functions (plus x0)." [1]
> >
> > As mentioned in my review, while the statement that this changes the dynamics of the system is an interesting one, this paper merely shows that the dynamics are different and that the residuals are important, rather than providing useful insights as to *how* they are different and what this means.
> >
> > As such, I reaffirm my initial score.
> >
> > [1] He K, Zhang X, Ren S, Sun J. Identity mappings in deep residual networks. InEuropean conference on computer vision 2016 Oct 8 (pp. 630-645). Springer, Cham.

---

### Official Review · AnonReviewer4 · 2018-11-08
**Technical contribution and readability needs to be improved given the  niche subject**

**Rating:** 4
**Confidence:** 4

**Review:**

In this paper the authors analyse the role of residuals in the performance of residual networks (res-nets) with shared layer weights and propose a method to adapt the depth of such networks during training. They exploit  the fact that res-nets identical blocks / shared layer weights are discrete time dynamical systems to conduct their analysis. Their main contribution seems to be an empirical evaluation of the role of transient dynamics in the performance of such res-nets. Experiments are done on a toy dataset on concentric circles and MNIST data.

I find the scope of the paper very narrow (res-nets with shared weights) even though there seem to be quite some other papers addressing this problem.

Clarity and quality of writing. It seems to me that the paper could  be much better written, the authors present long trains of thoughts and sometimes unsubstantiated arguments and at times there seems to be no clear storyline. In particular I would strongly  suggest a rewrite of  Sections 1, 2.1, and 4.  The storyline often very sketchy and is littered with hypothetical claims or less relevant information that can distracts and confuse the reader. I find that the hypotheses are not well formulated,   the claimed contributions of the paper don't seem to be significant enough and they should also be better connected to the experimental sections. Sometimes there are some odd choices of concepts/words such as "softmax algorithm" and "classification behaviour."

Technical quality and contribution. The main technical contribution of the paper seems to be the observation that  that the solution of an  ODE if the integral of the RHS an that one can derive an ODE for the residuals only. I don't find  these results significant/relevant enough to justify the publication of this paper.  I expected a better written and more informative Section 2.1: some approximations of rates of convergence, a bit more about basins of attraction. I am also a bit skeptical about the claim that the analysed network us a prey-predator model.

Experimental section. I find the description of experiments in Sec 4 very hard to read. The metrics are not clearly defined (not sure visualisations serve the purpose either), and the performed experiments are not well motivated and explained, for example in Section 4/P2 while I think I understand what the authors want to show (path of relevant neurons), I find the purpose of the whole experiment not very relevant.  I better analysis of the number of fixed points for classification tasks should be added, comparison of resulting features to other methods such as simple MLPs with same last layer could help . More relevant datasets should be be added e.g. CIFARxxx., ImageNet.

Overall: I find that this paper needs to be improved both in terms of readability and technical contribution to ready for publication.

---

> ### Author Response · Authors · 2018-11-27
> **Author response**
>
> We indeed thank the reviewer for his/her constructive and helpful comments.
>
> In the new version of the paper, we have added more analysis on the noise response of the network, using the dynamical system framework that we had before. Also, we improved the experiments.
> In fact, the experiments aimed at showing that the transient dynamics are indeed different and separable for different input classes. We have added more figures to explain this (in the appendix), which elaborate on the same concept as in figure 3.
>
> We found the suggestion of comparing ResNet with an MLP network very interesting, and we included that in the paper. We have shown that the transient dynamics in this case are oscillatory, and also fundamentally different than those of ResNets.
>
> The purpose of this study was to show the role of transient dynamics in input classification for a simple ResNet with the same number of neurons at each layer. Datasets such as CIFAR or ImageNet require dimensionality changes along the way of transient dynamics. This makes the analysis more difficult, and introduces changes in the autonomous dynamics. However, using these datasets is a subject of our future study.

---

### Meta-Review · Area_Chair1 · 2018-12-12
**novel yet unripe**

**Confidence:** 5
**Recommendation:** Reject

**Metareview:**

The paper uses dynamical systems theory to evaluate feed-forward neural networks.  The theory is used to compute the optimal depth of resnets.  An interesting approach, and a good initiative.

At the same time, the approach seems not to be thought through well enough, and the work needs another level of maturation before publication.  The application that is realised is too immature, and the corresponding contributions are not significant in their current form.  All reviewers agree on rejection of the paper.